# Efficacy and Treatment Satisfaction of Different Systemic Therapies in Children and Adolescents with Moderate-to-Severe Atopic Dermatitis: A Real-World Study

**DOI:** 10.3390/jcm12031175

**Published:** 2023-02-01

**Authors:** Sebastian Kiefer, Anke König, Viviane Gerger, Christine Rummenigge, Anne Christine Müller, Thomas Jung, Alexandra Frank, Georgios Tassopoulos, Emilie Laurent, Roland Kaufmann, Andreas Pinter

**Affiliations:** Department of Dermatology, Venereology and Allergology, University Hospital of Frankfurt, 60590 Frankfurt am Main, Germany

**Keywords:** atopic dermatitis, children, adolescents, dupilumab, upadacitinib, real-world data

## Abstract

For the treatment of moderate-to-severe atopic dermatitis in children and adolescents, the monoclonal antibody dupilumab and the selective JAK-1 inhibitor upadacitinib are two modern systemic therapies approved for long-term treatment. Both drugs have demonstrated high efficacy in randomized controlled trials, although evidence from real-world data in the pediatric population is limited. In a prospective analysis over 24 weeks, we investigated the efficacy, safety and treatment satisfaction of both systemic therapies in 23 patients (16 patients treated with dupilumab; 7 patients treated with upadacitinib). The median age of the patients was 16 years, with a median EASI of 18.8. A significant improvement in the EASI, VAS-itch, CDLQI, POEM and DFIQ from baseline to week 24 was demonstrated for both treatment options. No significant difference was observed between dupilumab and upadacitinib in any of the assessed scores. Less adverse events were recorded in the real-world setting compared with clinical trials. Our results confirm the efficacy and safety of dupilumab and upadacitinib as equivalent treatment options in children and adolescents in a real-world setting.

## 1. Introduction

Atopic dermatitis (AD) is a skin disease that typically presents as recurrent eczema and itching due to chronic inflammation, and it has a prevalence of up to 20% in the pediatric population [1]. There are numerous reasons leading to the development of atopic dermatitis, including a genetic predisposition in most cases. If both parents are affected, the child’s risk of developing AD is increased five-fold [2]. Atopic dermatitis has a negative impact on the quality of life of children and their families [3]. The disease is triggered by the interaction of various factors, such as a disturbed epidermal barrier, immune dysfunction, microbial imbalance and environmental factors [4]. Several inflammatory signaling pathways are pathophysiologically involved in the development of atopic dermatitis. In particular, a Th-2-cell-dominated immune response occurs, and many proinflammatory cytokines, such as IL-4, IL-5, IL-13, IL-17 and IL-31, are upregulated [5]. In addition to the use of emollients, the current treatment options for pediatric patients include topical corticosteroids, calcineurin inhibitors and UV therapy, as well as newer systemic therapies [6,7,8]. Since 2017, the human monoclonal antibody dupilumab has been approved in Europe for adults, and also for children aged six years and older since 2020. It specifically targets the IL-4-receptor-alpha-chain and thus mainly inhibits the effects of IL-4 and IL-13 [9,10,11]. Since 2022, the IL-13 cytokine inhibitor tralokinumab has been approved in patients 12 years of age and older [12]. A broader therapeutic approach involves the inhibition of the JAK-STAT pathway. The family of Janus kinases includes four subgroups (JAK-1, JAK-2, JAK-3 and TYK-2) responsible for modulating numerous inflammatory signaling pathways [5]. Since 2021, the selective JAK-1 inhibitor upadacitinib has been approved by the European Medicines Agency (EMA) for the treatment of atopic dermatitis in adolescents aged 12 years and older [13]. Thus, some modern systemic therapies are currently available for children and adolescents. Their efficacy and safety have been demonstrated in randomized trials [10,11,13,14,15]. Increasingly, real-world studies on the efficacy and safety of dupilumab are available [16,17,18]; however, comparable data concerning upadacitinib, particularly in a pediatric population, remain limited [19,20,21]. Moreover, there is still a lack of head-to-head studies between dupilumab and upadacitinib in this population. The aim of this prospective study is to provide real-world data regarding the efficacy and safety, as well as the treatment satisfaction, of dupilumab and upadacitinib in children and adolescents with AD.

## 2. Materials and Methods

### 2.1. Study Design and Population

In this prospective, open-label, non-interventional, monocentric study, children and adolescents aged 6 to 17 years suffering from moderate-to-severe atopic dermatitis were included. Data collection was performed from July 2021 to September 2022 at the Department of Dermatology, Venereology and Allergology, University Hospital Frankfurt, Germany. The treatment strategy was chosen in clinical routine independently of study participation. The patients and their parents were informed about the possibility of participating in this prospective study only after the treatment decision was made. A total of 33 patients were enrolled. Twenty patients received therapy with dupilumab, seven patients were treated with upadacitinib, five patients were treated with only a topical therapy and one patient was treated with methotrexate. In the present work, the data of those patients who received either dupilumab or upadacitinib are analyzed (*n* = 27). Four patients were excluded due to an incomplete data set (Figure 1).

The washout period for the systemic therapies was 2 weeks. The topical therapies were allowed to continue and a washout period was not deemed necessary. The dupilumab was dosed according to the approval status based on the patient’s body weight and age: 6 to 11-year-old children between 15 and 60 kg were treated with 300 mg dupilumab on day 0, day 15 and every 4 weeks thereafter. Above a weight of 60 kg, the initial dose was 600 mg, followed by 300 mg every 2 weeks. Adolescents between 12 and 17 years of age and under 60 kg received an initial dose of 400 mg, followed by 200 mg every other week. Adolescents weighing 60 kg or more were treated with an initial dose of 600 mg, followed by 300 mg every 2 weeks. The upadacitinib was used at a 15 mg dose, reflecting the approved dosage for adolescents. Children were not treated with upadacitinib because it is only approved from the age of 12 years. Depending on clinical need, additional topical corticosteroids (of mild-to-medium potency) and topical calcineurin inhibitors were allowed. No additional UV therapy was used by any patient. The daily use of emollients was recommended to all the patients. Follow-up visits were scheduled 6, 12 and 24 weeks after the initiation of therapy. The study was conducted in accordance with the Declaration of Helsinki and approved by the Ethics Committee of the Department of Medicine at Goethe University Hospital (N: 2021-24).

### 2.2. Data Collection

The baseline data included: sex, age, BMI, mean age of onset, prior therapies, family history regarding atopic dermatitis, comorbidities and allergies. The Eczema Area and Severity Index (EASI) was also assessed at each visit. Itching and treatment satisfaction were measured using a visual analog scale ranging from 0 to 10 cm (VAS-itch and VAS-treatment satisfaction). The subjective impact of eczema on the quality of life was assessed using the Children’s Dermatology Life Quality Index (CDLQI). An evaluation of the atopic dermatitis severity by the parents was carried out via the Patient-Oriented Eczema Measure (POEM). The impact of the child’s disease on the family was assessed by the parents using the Dermatitis Family Impact Questionnaire (DFIQ).

### 2.3. Statistics

A descriptive analysis of the study population and the two subgroups (dupilumab treatment versus upadacitinib treatment) was performed to present the relevant demographic and clinical characteristics. The continuous variables were presented as the median with the interquartile range, while the categorical variables were presented as an absolute number with the percentage, as appropriate. The Mann–Whitney U test was used to compare the two independent subgroups when the data set was not normally distributed, and the Wilcoxon signed-rank test was used to analyze the dependent variables during the course of the therapy, comparing the baseline with week 6, week 12 and week 24. We considered a *p*-value < 0.05 to be statistically significant. All the calculations were performed using the statistical program JASP, version 0.16.2 (JASP Team, Amsterdam, The Netherlands), for Windows.

## 3. Results

### 3.1. Study Population

In total, the study population included 23 patients. The baseline data are shown in Table 1. The median age was 16 years. A total of 52.2% (*n* = 12) of patients were male and 47.8% (*n* = 11) were female. More adolescents than children were enrolled in the study: 82.6% (*n* = 19) of study participants were between 12 and 17 years of age and 17.4% (*n* = 4) were between 6 and 11 years of age. The median age at the first manifestation of atopic dermatitis was three years. The most common concomitant atopic disease was rhinoconjunctivitis in 34.8% (*n* = 8) of patients, followed by allergic asthma in 30.4% (*n* = 7) and food allergies in 26.1% (*n* = 6) of all patients. Overall, 69.6% (*n* = 16) had at least one allergy in their medical history. Pollen allergy (43.5%, *n* = 10) and house dust mite allergy (34.8%, *n* = 8) were most frequently reported. A positive family history of atopic dermatitis was present in 43.5% (*n* = 10) of patients. Almost all the patients had previously used topical therapy (91.3%, *n* = 21). Medium-potency steroids (78.3%, *n* = 18) were used most frequently and high-potency steroids (30.4%, *n* = 7) less frequently. A total of 56.5% (*n* = 13) of patients had undergone prior treatment with calcineurin inhibitors. No other topical treatments were previously used. Previous systemic treatment had been received by 26.1% (*n* = 6) of patients: three with prednisolone and three with dupilumab.

The patients suffered from severe atopic dermatitis at baseline, with a median EASI score of 18.8 and an itch score of 9.0 cm on the VAS-itch. The patients’ quality of life was generally impaired; the median value of the CDLQI was 16 points. The atopic dermatitis was rated as very severe by the parents at baseline (median POEM score of 25.0) and said to have a major impact on the family’s quality of life (median DFIQ score of 12.0).

### 3.2. Baseline Data of the Subgroups

The baseline data of the two treatment groups did not differ significantly (Table 2). The median EASI score was not significantly lower in the dupilumab group than in the upadacitinib group (18.1 vs. 22.0, *p* > 0.05). There was also no significant difference in terms of the itch score (8.9 vs. 9.6, *p* > 0.05), impact of disease on quality of life and subjective disease severity (CDLQI 17.7 vs. 12.0, POEM 25.5 vs. 24.0, DFIQ 12.0 vs. 9.0, each *p* > 0.05). Regarding the demographic data (sex, age, BMI), the distribution was balanced in both therapy groups.

### 3.3. Efficacy of Systemic Therapy

Both therapies significantly improved the eczema as well as the itch and quality-of-life scores. The greatest improvement in all the assessed scores was achieved within the first 6 weeks of therapy (Table 3).

The median EASI score was significantly reduced from 18.8 at baseline to 9.0 at week 6, 5.7 at week 12 and 6.9 at week 24 (*p* < 0.001; Table 3). At least a 50% decrease in the Eczema Area and Severity Index (EASI 50) was achieved by 65.2% of patients at week 6, 78.3% at week 12 and 82.6% at week 24. An EASI score of 75 was reached by 13.0% of patients at week 6, 34.8% at week 12 and 43.5% at week 24 (Figure 2).

The therapy also significantly reduced the patients’ itching: the median VAS-itch score decreased from 9.0 at baseline to 5.0 at week 6 and 4.0 at weeks 12 and 24 (*p* < 0.001; Table 3). An itch reduction of at least 4 cm on the visual analog scale was achieved by 39.1%, 52.2% and 56.5% of patients at weeks 6, 12 and 24, respectively (Figure 3).

The improvements in their skin and itching were associated with an improvement in the patients’ quality of life. The median CDLQI score was more than halved throughout the observation period (baseline 16.0 vs. week 24 6.0). A reduction in the CDLQI and POEM score of 6 or more points is considered clinically meaningful [22,23]. At week 24, this was achieved by 69.6% (*n* = 16) in the CDLQI score and by 87% (*n* = 20) in the POEM score. As the children’s and adolescents’ atopic dermatitis improved, the impact of the disease on their family also continuously decreased. Accordingly, the median DFIQ score declined from a baseline score of 12 points to 3 points at week 24 (*p* < 0.01; Table 3). We observed no difference between the male and female patients regarding the efficacy and safety outcomes.

### 3.4. Treatment Satisfaction

The median baseline VAS-treatment satisfaction score of 2.5 was significantly increased to 7.1 at weeks 6 and 12 and to 7.5 at week 24 (*p* < 0.001; Table 3). Again, the greatest change occurred during the first 6 weeks of therapy.

### 3.5. Comparison of Efficacy and Safety between Dupilumab and Upadacitinib

No statistically significant difference was observed in any of the assessed scores between the dupilumab and upadacitinib therapy throughout the observation period (Figure 4, Figure 5 and Figure 6). From baseline to week 6, the median EASI score decreased more numerically, albeit not in a statistically significant manner, in the upadacitinib group compared to the dupilumab group (22 to 7.6 vs. 18.1 to 9.5; Figure 4). The lowest disease activity was achieved at week 12 in both therapy groups (5.6 vs. 5.7). The EASI score at week 24 was again slightly lower in the dupilumab group than in the upadacitinib group (5.8 vs. 7.7), and slightly worse overall compared with week 12.

A numerically faster and not statistically significant reduction in itching was also observed at the beginning of the therapy in the upadacitinib group compared with the dupilumab group (9.6 to 4.8 vs. 8.9 to 5.1 at week 6; Figure 5). However, in terms of the skin improvement and itch reduction, the treatment efficacy converged again in the long-term comparison.

The quality-of-life scores also showed no significant difference between the two therapies. The CDLQI score and the DFIQ score were similarly effectively improved in both subgroups (Figure 6). The POEM score indicated the faster treatment response of upadacitinib, as also seen in relation to the EASI and VAS-itch scores, albeit without showing a statistically significant difference. Regarding the subjective treatment satisfaction of the patients, both therapies were equivalent. The intensity and duration of the additive topical treatment did not differ relevantly between the two treatment groups. Most patients no longer required topical treatment by week 6.

### 3.6. Safety

Five of the twenty-three patients (21.8%) reported at least one adverse event. These side effects were mild and did not result in therapy interruption, dose modification or discontinuation. During the dupilumab treatment, two patients developed conjunctivitis (grades 1–2), one patient developed herpes labialis (grade 2) and one patient developed mild psoriasis vulgaris. Among the patients treated with upadacitinib, one patient suffered from a mild upper respiratory tract infection.

## 4. Discussion

Both dupilumab and upadacitinib have already shown good efficacy in the treatment of atopic dermatitis in randomized controlled trials [9,10,11,14,15]. Our real-world study confirms that these modern systemic therapies are an effective treatment for atopic dermatitis in children and adolescents.

The EASI score was significantly reduced during the treatment. At week 24, EASI scores of 50 and 75 were achieved by 83% and 44% of all patients, respectively. Simpson et al. described an EASI 75 response of 38.1% (300 mg dupilumab every 4 weeks) in the pivotal randomized controlled trial of dupilumab in adolescents [10]. Our results are comparable to this. Napolitano et al. and Hagino et al. described higher EASI 75 scores in real-world studies on the efficacy of dupilumab or upadacitinib in children and adolescents [17,18,21]. The possible reasons for the moderate EASI response in our study include the selected, and in many cases already pretreated (91%), patient population in a tertiary center. Nevertheless, the treatment satisfaction of our patients was high (VAS-treatment satisfaction at baseline 2.5 vs. 7.5 at week 24). This may be mainly explained by the strong reduction in itching, with not only the improvement in the skin but particularly the reduction in itching being decisive for patients’ therapy satisfaction [24]. The median baseline value of 9.0 cm on the VAS-itch was more than halved to 4.0 cm in week 24 (*p* < 0.001). A total of 58% of patients achieved an improvement of ≥4 cm on the VAS-itch at week 24. Thus, our patients achieved an itch reduction comparable to that seen in the randomized controlled pivotal trials of dupilumab and upadacitinib [9,10,11,14]. Additionally, the improvement in both skin and itching was associated with gains in quality of life for the children and adolescents (baseline CDLQI 16 vs. week 24 CDLQI 6) and their families (baseline DFIQ 12 vs. week 24 DFIQ 3). The positive correlation between atopic dermatitis severity and the influence on the quality of life of the patients and their families has been described many times [25,26].

When comparing the treatment efficacy of dupilumab versus upadacitinib, we did not find a significant difference. This applies to the improvement in the EASI, as well as in the VAS-itch and quality of life scores. The disease severity at baseline was not significantly different in the two treatment groups (EASI 18.1 vs. EASI 22.0, *p* > 0.05 and VAS-itch 8.9 vs. 9.6, *p* > 0.05), and both treatments were equally available in Germany. The possible bias due to the lack of randomization can be assumed to be low. The faster treatment response of upadacitinib described in the literature was also partially reflected in our data [14,27]. The EASI score at week 6 decreased more rapidly in the upadacitinib group than in the dupilumab group, although the effect was not statistically significant. The significant effect of the JAK-1 inhibitor might be seen at earlier time points. Currently, there are no published head-to-head studies of dupilumab and upadacitinib in a pediatric patient population. Blauvelt et al. demonstrated the superiority of 30 mg upadacitinib daily versus 300 mg dupilumab every 2 weeks in adults [28]. Data from a meta-analysis by Silverberg et al. suggest that the efficacy of 15 mg upadacitinib daily is comparable to that of 300 mg dupilumab every 2 weeks and superior to that of 200 mg dupilumab every 2 weeks [29]. In line with this, we did not find a significant difference between the 15 mg upadacitinib and dupilumab in our study.

All these results have to be judged in the context of the following limitations. The study population included only 23 patients. No distinction was made between the different dupilumab doses, and no blinded assessment was performed. In addition, the study was monocentric.

## 5. Conclusions

Our results support the notion that children and adolescents with moderate-to-severe atopic dermatitis benefit from systemic treatment, which can significantly improve the severity of eczema, the intensity of itching and the quality of life (patients and relatives). In this small cohort, dupilumab and upadacitinib had similar effects in an adolescent population. Both therapies have a good safety profile. As there seems to be no clearly superior therapy, it is even more important that individual patient counseling precedes a treatment decision. It is desirable that, in the future, further real-world studies, as well as randomized controlled head-to-head studies, on the efficacy of modern systemic therapies in children and adolescents with moderate-to-severe atopic dermatitis be conducted.

## Figures and Tables

**Figure 1 jcm-12-01175-f001:**
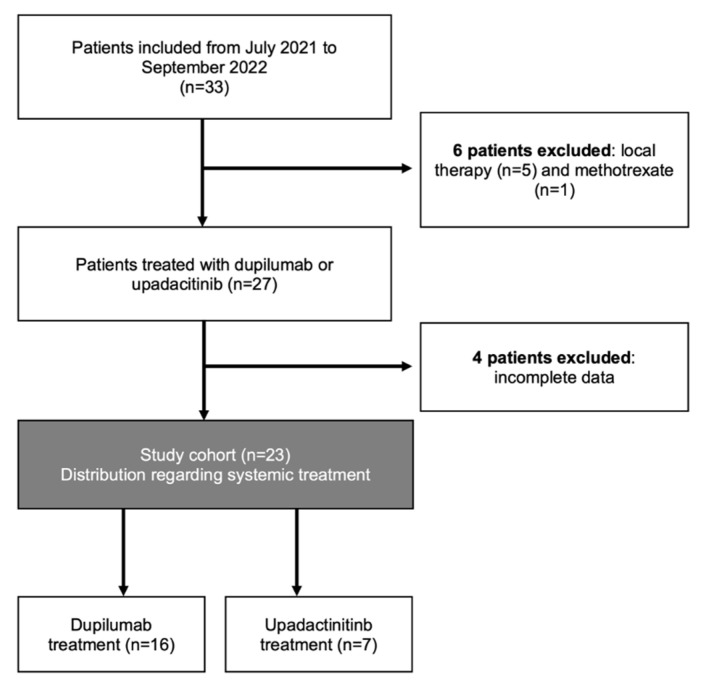
Study cohort.

**Figure 2 jcm-12-01175-f002:**
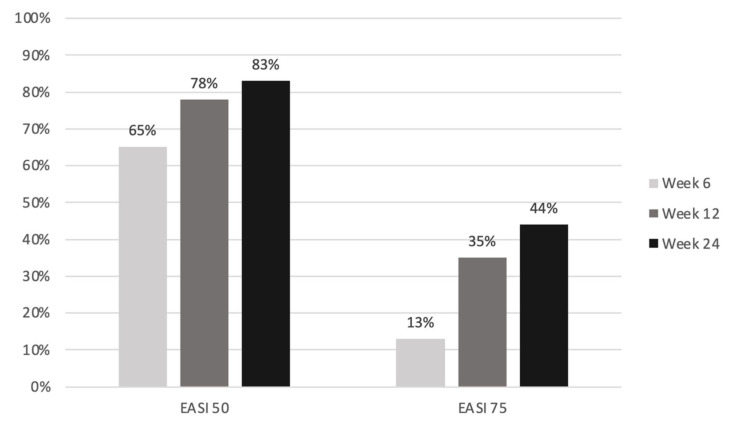
Percentage of patients with a 50% (EASI 50) or 75% (EASI 75) improvement in EASI.

**Figure 3 jcm-12-01175-f003:**
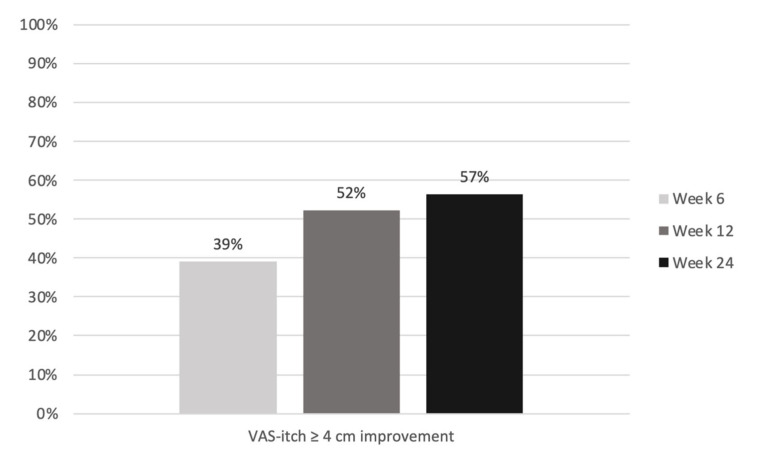
Percentage of patients with a clinically significant improvement of at least 4 cm in the VAS-itch.

**Figure 4 jcm-12-01175-f004:**
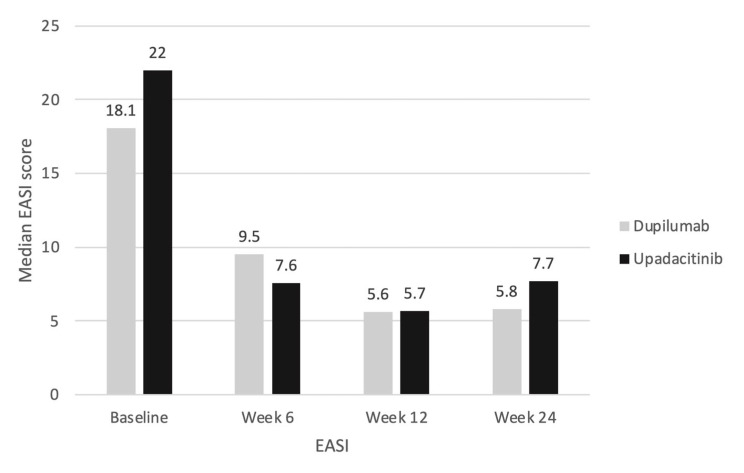
Median EASI score; *p*-value in Mann–Whitney U test for dupilumab (*n* = 16) vs. upadacitinib (*n* = 7) subgroup comparison (all time *p* > 0.05).

**Figure 5 jcm-12-01175-f005:**
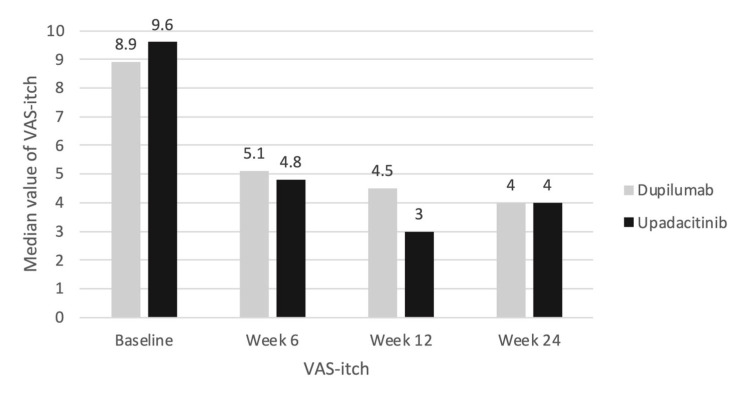
Median VAS-itch score; *p*-value in Mann–Whitney U test for dupilumab (*n* = 16) vs. upadacitinib (*n* = 7) subgroup comparison (all time *p* > 0.05).

**Figure 6 jcm-12-01175-f006:**
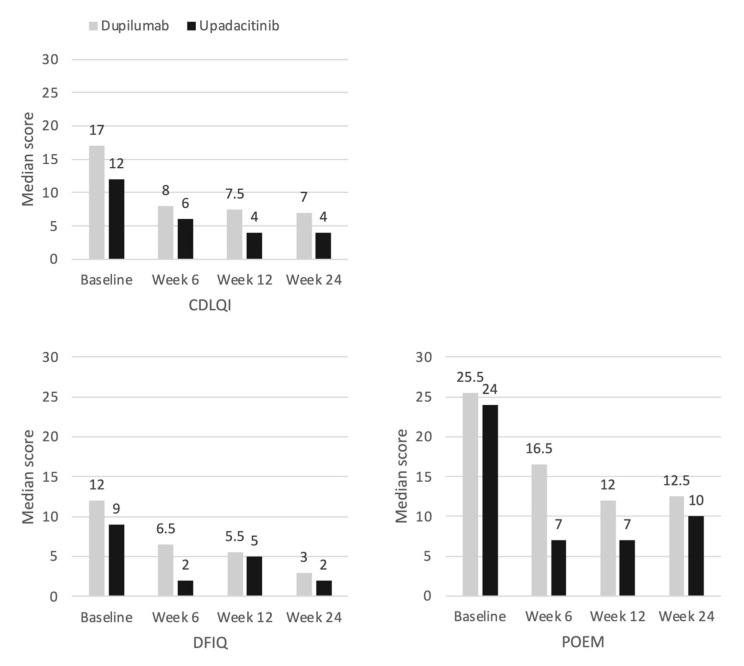
Median value of CDLQI, POEM and DFIQ; *p*-value in Mann–Whitney U test for dupilumab (*n* = 16) vs. upadacitinib (*n* = 7) subgroup comparison (all time *p* > 0.05).

**Table 1 jcm-12-01175-t001:** Baseline demographics and clinical characteristics.

	Study Cohort (*n* = 23)
**Sex, *n* (%)**	
Male	12 (52.2)
Female	11 (47.8)
**BMI, median (IQR)**	20.7 (16.8–23.9)
**Age in years, median (IQR), [range]**	16.0 (12.5–17.0) [6–17]
**Age groups, *n* (%)**	
6–11 years	4 (17.4)
12–17 years	19 (82.6)
**Age at first manifestation in years, median (IQR)**	3.0 (0.5–5.0)
**Concomitant diseases, *n* (%)**	
Asthma	7 (30.4)
Food allergies	6 (26.1)
Rhinoconjunctivitis	8 (34.8)
**Allergies,** ** *n* ** **(%)**	16 (69.6)
Pollen allergy	10 (43.5)
House dust mite allergy	8 (34.8)
**Previous therapies, *n* (%)**	
Yes	21 (91.3)
No	2 (8.7)
**Topical therapy, *n* (%)**	
Low-potency steroids	3 (13.0)
Medium-potency steroids	18 (78.3)
High-potency steroids	7 (30.4)
Ultra-high-potency steroids	3 (13.0)
Calcineurin inhibitors	13 (56.5)
**Previous systemic therapies, *n* (%)**	6 (26.1)
Prednisolone	3 (13.0)
Dupilumab	3 (13.0)
**Positive family history of AD, *n* (%)**	10 (43.5)
**Objective outcome measure, median (IQR)**	
EASI	18.8 (15.4–25.5)
**Subjective outcome measures, median (IQR)**	
VAS-itch	9.0 (8.0–9.7)
POEM	25.0 (22.0–27.5)
CDLQI	16.0 (11.0–19.0)
DFIQ	12.0 (6.5–13.5)

IQR = interquartile range; BMI = Body Mass Index; EASI = Eczema Area and Severity Index; VAS = Visual Analogue Scale; AD = atopic dermatitis; POEM= Patient-Oriented Eczema Measure; CDLQI = Children’s Dermatology Life Quality Index; DFIQ = Dermatitis Family Impact Questionnaire.

**Table 2 jcm-12-01175-t002:** Baseline demographics and clinical characteristics of the subgroups.

	Dupilumab (*n* = 16)	Upadacitinib (*n* = 7)
**Sex, *n* (%)**		
Male	8 (50.0)	4 (47.1)
Female	8 (50.0)	3 (42.9)
**BMI, median (IQR)**	19.8 (15.6–23.6)	21.1 (19.4–24.4)
**Age, median (IQR), [range]**	16.0 (12.0–17.0) [6–17]	16.0 (14.0–16.0) [12–17]
**Previous therapies, *n* (%)**		
Yes	15 (93.8)	6 (85.7)
No	1 (6.2)	1 (14.3)
**Topical therapy, *n* (%)**		
Low-potency steroids	3 (18.8)	0 (0)
Medium-potency steroids	12 (75.0)	6 (85.7)
High-potency steroids	5 (31.3)	2 (28.6)
Ultra-high-potency steroids	3 (18.8)	0 (0)
Calcineurin inhibitors	11 (68.8)	2 (28.6)
**Previous systemic therapies, *n* (%)**	4 (25.0)	2 (28.6)
Prednisolone	3 (18.8)	0 (0)
Dupilumab	1 (6.3)	2 (28.6)
**Objective outcome measure, median (IQR)**		
EASI *	18.1 (14.3–28.7)	22.0 (18.7–23.3)
**Subjective outcome measures** **, median (IQR)**		
VAS-itch *	8.9 (8.0–9.4)	9.6 (8.5–10.0)
POEM *	25.5 (21.0–27.3)	24.0 (22.0–26.5)
CDLQI *	17.0 (11.8–19.0)	12.0 (10.0–18.0)
DFIQ *	12.0 (7.5–12.5)	9.0 (6.0–14.0)

* *p*-value in Mann–Whitney U test for dupilumab vs. upadacitinib subgroup comparison > 0.05. IQR = interquartile range; BMI = Body Mass Index; EASI = Eczema Area and Severity Index; VAS = Visual Analogue Scale; POEM = Patient-Oriented Eczema Measure; CDLQI = Children’s Dermatology Life Quality Index; DFIQ = Dermatitis Family Impact Questionnaire.

**Table 3 jcm-12-01175-t003:** Treatment efficacy in all patients presented as median scores (with interquartile range).

	Baseline	Week 6	Week 12	Week 24	*p*-Value ***
EASI	18.8 (15.4–25.5)	9.0 (5.5–11.7)	5.7 (3.5–10.0)	6.9 (2.7–8.8)	<0.001
VAS-itch	9.0 (8.0–9.7)	5.0 (3.2–6.5)	4.0 (1.9–6.8)	4.0 (2.0–6.6)	<0.001
VAS-treatment satisfaction	2.5 (0.9–4.5)	7.1 (5.8–8.8)	7.1 (5.6–9.3)	7.5 (6.0–8.7)	<0.001
CDLQI	16.0 (11.0–19.0)	7.0 (5.0–12.0)	6.0 (3.5–10.5)	6.0 (3.5–10.5)	<0.001
DFIQ	12.0 (6.5–13.5)	6.0 (3.0–8.5)	5.0 (2.0–7.5)	3.0 (1.0–4.0)	<0.01
POEM	25.0 (22.0–27.5)	12.0 (7.0–18.0)	10.0 (7.0–14.5)	12.0 (6.5–16.0)	<0.001

* *p*-value in Wilcoxon signed-rank test for the comparison at baseline, respectively, at weeks 6, 12 and 24.; EASI = Eczema Area and Severity Index; VAS = Visual Analogue Scale; POEM = Patient-Oriented Eczema Measure; CDLQI = Children’s Dermatology Life Quality Index; DFIQ = Dermatitis Family Impact Questionnaire.

## Data Availability

Not applicable.

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
