# Peer review of "Efficacy and Treatment Satisfaction of Different Systemic Therapies in Children and Adolescents with Moderate-to-Severe Atopic Dermatitis: A Real-World Study"

_jcm, 2023, doi:10.3390/jcm12031175_

Round 1
Reviewer 1 Report
The manuscript by Sebastian et al. describes the effects of monoclonal antibody dupilumab and the selective JAK1 inhibitor upadacitinib in children and adolescents with moderate to severe atopic dermatitis in the real world. This group was able to record and describe the eczema status before and after treatment with the two drugs in a clear way. Similar significant improvements in EASI, VAS-itch, CDLQI, POEM and DFIQ after treatment demonstrated that no significant difference was observed between dupilumab and upadacitinib in the real-world study. Overall, the paper is well written, and the study is well designed. It would be perfect if the author could recruit more patients for the study.
Reviewer 2 Report
This study is interesting and potentially would be useful if there were more patients, if the treatments were randomized, and if there were more childhood patients, specifically. Because of inherent bias in prescribing medication (outside of a randomized trial) it is hard to compare treatments, and hard to generalize this to clinical practice. Given the already low number of patients, and the fact that no children received one of the interventions, and only 4 children were included, please consider removing the childhood cases and reporting the adolescent comparison.
“The treatment strategy was chosen in clinical routine independently 66 of study participation. ” However treatments were not randomized and there is a selection bias — perhaps more severe patients receiving one or the other medication, or perhaps for some reason all of the options were not available for all patients (i.e. in my practice it would be hard to get upadacitinib covered when dupilumab is supported by approval and insurances). This needs to be addressed in the discussion.
The two interventions reported here, dupilumab and upadacitinib should be profiled separately in Table 1 - they are two separate study cohorts.
In 2022 there are many additional topical treatments available that are relevant to report (as prior treatments) or acknowledge if not utilized: eucrisa/crisaborole ointment and opzelura cream are effective for atopic dermatitis. Were these not used? If not, please acknowledge.
It appears that no patients <12 years were treated with upadacitinib; it is helpful to consider the response of pre-adolescent children and report on any differences noted in this population (for dupilumab) and state the lack of children that received upadacitinib.
Conclusions are a bit strong for this small cohort- please revise to:
Our results support that children and adolescents with moderate-to-severe atopic dermatitis benefit from a systemic treatment.
With such a low sample, it is not fair to state that “Dupilumab and updacitinib appear to be equivalent therapeutic options” without qualifying…. I recommend to revise to: “in this small cohort, dupilumab and updacitinib had similar effect in an adolescent population.”
Reviewer 3 Report
Comments: Authors reported comparison of efficacy and safety of dupilumab and upadacitinib in children and adolescents with moderate-to-severe atopic dermatitis that will improve the understanding these therapies. The manuscript requires consideration on following comments:
1) There are many studies available reporting similar outcome with dupilumab on larger sample sizechildren and adolescents, however the study was dominated with adolescent. A separate comparison of adolescent and children group should be provided.
Have authors
4) Dosage of Dupilumab should be clearly defined. Authors mentioned in Table#1 that majority of the patients were on therapy previously. Please clearly state the selection criteria, how patients were transferred from previous therapy to present dosage of the study. Is there any drug wash out of previous treatment applied or were maintained on some topical therapy? Detail explanation should be incorporated in the study.
5) Please discuss the findings of the recent Upadacitinib therapy reported by Simpson et al. (reference# 11 in the present manuscript). Is there any difference observed in terms of outcome of the two studies.
6) It is highly recommended to discuss the related findings in detail and compare the outcome including the adverse events. Only few studies were discussed in detail in the discussion section.
7) The major limitation of the present study was small sample size and lack of control arm (placebo effect) that practically limited the conclusive outcome. Authors should take care of same in their future studies.
Reviewer 4 Report
The authors reported the results of a real-world study investigating the effectiveness and treatment satisfaction of different systemic therapies in children and adolescents with moderate-to-severe atopic dermatitis. The manuscript is interesting and well-written. However, I have some suggestions.
My comments:
- English language should be revised;
- Abstract: before use abbreviations (EASI,..) you should report the words you are abbreviating;
Abstract: the values of EASI, VAS-itch, CDLQI, POEM and DFIQ at baseline and at each timepoint shpuld be reported
- Introduction: "Since 2017, the human monoclonal antibody dupilumab has been 42 approved in Europe for adults and since 2020 also for children aged 6 years and older" REF?
- Introduction: more general data about the effectiveness and safety of dupilumab and upadacitinib in real-life should be discussed in order to increase the background. You should read and cite 10.1007/s40257-021-00597-5 ; 10.1080/09546634.2022.2102121 ; 10.1080/09546634.2019.1682503 ; 10.1016/j.jaad.2020.08.127 ; 10.1111/dth.15120 ; 10.1111/jdv.18311; 10.1007/s13555-022-00882-z.
- Material and Methods: inclusion and exclusion criteria should be reported
- Material and Methods: please specify dupilumab dosage
- Results: you should compare the efficacy and safety of dupilumab and upadacitinib only for patients of the same age groups...
- Discussion: I appreciated the review of the current literature regarding the study on the efficacy and safety of the investigated drugs in the adolescents. However, you should compare your results with theirs. Moreover, a table summarizing current studies should be interesting.
- Strengths and limitations: strengths and limitations should be discussed in a separate section
- Table and figures: a table summarizing current studies should be interesting as discussed before.
Round 2
Reviewer 4 Report
All the changes have been made. The manuscript is now suitable for publication.